# Histological Characterization of Class I HLA Molecules in Whole Umbilical Cord Tissue Towards an Inexhaustible Graft Alternative for Reconstructive Surgery

**DOI:** 10.3390/bioengineering10010110

**Published:** 2023-01-12

**Authors:** Yue Ying Yao, Dennis K. Lee, Stephanie Jarvi, Marjan Farshadi, Minzhi Sheng, Sara Mar, Ori Nevo, Hon S. Leong

**Affiliations:** 1Department of Medical Biophysics, Temerty Faculty of Medicine, University of Toronto, Toronto, ON M5G 1L7, Canada; 2Biological Sciences Platform, Sunnybrook Research Institute, Toronto, ON M4N 3M5, Canada; 3Temerty Faculty of Medicine, University of Toronto, Toronto, ON M5S 1A8, Canada; 4Department of Obstetrics and Gynecology, Temerty Faculty of Medicine, University of Toronto, Toronto, ON M5S 1A8, Canada

**Keywords:** umbilical cord, HLA-G, HLA-ABC, HLA-E, histology, mass spectrometry

## Abstract

Background: Limited graft availability is a constant clinical concern. Hence, the umbilical cord (UC) is an attractive alternative to autologous grafts. The UC is an inexhaustible tissue source, and its removal is harmless and part of standard of care after the birth of the baby. Minimal information exists regarding the immunological profile of a whole UC when it is considered to be used as a tissue graft. We aimed to characterize the localization and levels of class I human leukocyte antigens (HLAs) to understand the allogenicity of the UC. Additionally, HLA-E and HLA-G are putative immunosuppressive antigens that are abundant in placenta, but their profiles in UC whole tissue are unclear. Hypothesis: The UC as a whole expresses a relatively low but ubiquitous level of HLA-ABC and significant levels of HLA-G and HLA-E. Methods: Healthy patients with no known pregnancy-related complications were approached for informed consent. UCs at term and between 12 and 19 weeks were collected to compare HLA profiles by gestational age. Formalin-fixed paraffin-embedded tissues were sectioned to 5 µm and immunohistochemically stained with a pan-HLA-ABC, two HLA-G-specific, or an HLA-E-specific antibody. Results: HLA-ABC was consistently found present in UCs. HLA-ABC was most concentrated in the UC vessel walls and amniotic epithelium but more dispersed in the Wharton’s Jelly. HLA-E had a similar localization pattern to HLA-ABC in whole UC tissues at both gestational ages, but its protein level was lower. HLA-G localization and intensity were poor in all UC tissues analyzed, but additional analyses by Western immunoblot and mass spectrometry revealed a low level of HLA-G in the UC. Conclusion: The UC may address limitations of graft availability. Rather than the presence of HLA-G, the immunosuppressive properties of the UC are more likely due to the abundance of HLA-E and the interaction known to occur between HLA-E and HLA–ABC. The co-localization of HLA-E and HLA-ABC suggests that HLA-E is likely presenting HLA-ABC leader peptides to immune cells, which is known to have a primarily inhibitory effect.

## 1. Introduction

One of the constant disadvantages of clinically using autologous tissue grafts is limited graft availability [1]. One example is in the surgical treatment of hypospadias, which is a common congenital anomaly in males where the urethral opening is abnormally located [2]. Currently, patients with severe hypospadias undergo urethral reconstructive surgery using autologous grafts, but when repeat operations or extensive grafting is needed, graft availability is an ongoing concern [3,4]. In recent years, the UC has been studied both as a source of mesenchymal stem cells (MSCs) and as allografts for therapeutic purposes [5,6,7,8,9,10,11,12,13,14,15]. The UC is conceivably an inexhaustible source of tissue with no risk of graft unavailability, as it is disposed as medical waste after birth and its removal is harmless. Furthermore, when used as a tissue graft for tendon regeneration in a rabbit model, the transplanted UC tissue differentiated to a tendon-like morphology in post-transplantation follow-ups [11]. In conjunction with reports of immunosuppressive properties of UC-derived cells, the UC is an attractive alternative to patient-derived grafts for reconstructive surgery.

The UC is an extraembryonic tissue that serves as a bidirectional transport system between the fetus and the placenta to support fetal development during pregnancy [16]. It is composed of an outer amniotic epithelium (AE) and three umbilical vessels surrounded by Wharton’s Jelly (WJ), which is a loose connective tissue that protects and provides structural support to the vessel [16,17]. In the past, the immunological profiles of cord lining epithelial cells (CLECs) and WJ-derived mesenchymal stem cells (WJMSCs) have been the most studied. Characterization studies of CLECs and WJMSCs reported no HLA-DR expression, but HLA-A, HLA-B, and HLA-C (HLA-ABC) expression was present [5,7,8,9,10,18,19,20,21,22,23,24,25]. Rejection of an allogeneic transplant occurs when the donor cells are recognized as foreign antigens by the recipient’s T cells and an immune response is launched [26]. The human classical major histocompatibility complex class-I (MHC-I) molecules HLA-ABC are among those that can be recognized by recipient T cells [26]. Despite HLA-ABC presence, co-cultures of CLEC or WJMSCs with activated allogeneic lymphocytes showed inhibition of lymphocyte proliferation [5,7,8,10,19,20,23,24,27]. In xenotransplants, CLECs were observed to reduce the rejection rate of neighboring allogeneic cells, and open wounds that received WJMSCs healed better and faster [5,10]. Clinically, intravenous injection of WJMSCs into patients with chronic heart failure induced no signs of rejection [7]. The immunosuppressive and wound healing properties of UC-derived cells lend support to the notion that UC allografts may be similarly low in immunogenicity.

The immunosuppressive properties of CLECs and UCMSCs have mainly been attributed to the presence of the non-classical MHC-I molecule HLA-G [5,7,10,19,20,23,24,27,28]. HLA-G is known to be crucial in the establishment and maintenance of maternal tolerance towards the semi-foreign fetus in pregnancy [23,29]. In allotransplantations, HLA-G presence can enhance tolerance and reduce rejection of the donor tissue [30]. Immunosuppressive actions of HLA-G include the direct suppression of immune cell proliferation and immune cytotoxic activity as well as indirectly through another non-classical MHC-I molecule, HLA-E [31]. This interaction inhibits natural killer (NK) cell cytotoxicity and has been suggested to inhibit uterine NK cell lysis, which contributes to maternal tolerance towards the fetus in pregnancy [32]. In addition, HLA-E has also been reported to present peptides derived from leader sequences of HLA-ABC, which has similar immunosuppressive effects on immune cell activity [33,34]. The presence and levels of HLA-G and HLA-E in whole UC tissue would lend support to its observed low immunogenicity in an allogeneic environment.

Despite growing interests in using the UC as tissue grafts, little evidence exists regarding the localization and levels of class-I HLA molecules in the context of the whole cord. Here, we investigate and compare the localization and levels of HLA-ABC, HLA-G, and HLA-E in whole-cord cross-sections to better understand the allogenicity of the UC.

## 2. Materials and Methods

### 2.1. Sample Collection

This study was approved by the Sunnybrook Health Sciences Centre Research Ethics Board (#136-2008), and all patients provided informed consent. Term placental and UC samples (*n* = 10, Table 1) were collected after elective caesarean sections at Sunnybrook Health Sciences Centre (Toronto, ON, Canada). All participating patients experienced a healthy pregnancy with no signs of high blood pressure, preeclampsia, gestational hypertension, gestational diabetes, infection, renal dysfunction, or existing conditions that would impact the pregnancy outcome or fetal development (Table 1). Gravidity is defined as the total number of times a patient has been pregnant and parity is defined as the number of times a patient has carried the pregnancy to at least 22 gestational weeks (Table 1). Placental and UC samples from early pregnancy were collected from elective pregnancy terminations of anonymous patients (*n* = 10, 12–19 weeks). For samples collected at term, the decidua and trophoblast villi were dissected as whole tissue blocks from the placenta, and the entire length of the UC was removed at 2 cm from its maternal end immediately following delivery. Decidual tissues and UC segments from early pregnancy were rinsed and collected in phosphate buffered saline (PBS). Tissue samples were gently washed in PBS to remove residual blood and submerged in PBS at 4 °C until fixation. Tissue samples were also dissected to small fragments, flash frozen in liquid nitrogen, and stored at −80 °C.

### 2.2. Immunohistochemistry

Immunohistochemistry (IHC) staining against HLA-ABC, HLA-G, and HLA-E were performed on UC tissue cross-sections that were collected from term pregnancies and between 12 and 19 weeks. Gestational-age-matched placental samples were used during the initial IHC protocol optimization process and as positive controls for all subsequent experiments. A mouse IgG1 antibody isotype (Clone G3A1; Cat 5415S, Cell Signaling, Danvers, MA, USA) was used as a negative control at the same working concentration as the corresponding primary antibodies. Table 2 lists all the primary antibodies used and their optimized working concentrations.

Tissues were dissected and fixed in 10% neutral buffered formalin (NBF) within 24 h of collection. To determine the optimal fixation conditions for the UC, UC samples were dissected to approximately 5 mm thick cross-sections and fixed for 24 or 48 h, with or without rotary agitation, and at 4 °C or room temperature. Upon comparing the HLA-ABC staining results of UC tissue cross-sections at term, the optimal fixation conditions were determined to be 48 h with rotary agitation at room temperature. Placental samples were fixed at 4 °C overnight with gentle rotary agitation, washed twice in PBS for 20 min each, post-fixed in 10% NBF for 30 min, and washed twice in PBS for 20 min, each at room temperature. All fixed tissue samples were submerged in 70% ethanol at 4 °C until paraffin embedding.

Formalin-fixed paraffin-embedded tissue samples were sectioned to 5 µm on Superfrost Plus slides (Electron Microscopy Sciences), with each slide containing two serial sections of one sample. Slides were baked at 55 °C for 1 h prior to IHC staining to enhance tissue adhesion to slides. Slides were deparaffinized in xylene, rehydrated in a decreasing gradient of ethanol in distilled water (dH_2_O), and washed twice in PBS for 5 min each. Five different antigen retrieval (AR) protocols were tested on placental and UC samples to optimize the tissue preservation and epitope exposure for each tissue type and antibody. Two heat-induced epitope retrieval (HIER) protocols were tested: microwave and pressure cooker (PC). With the microwave protocol, slides were placed in a slide container and completely submerged in 0.01 M sodium citrate buffer (pH 6). Then, the submerged slides were heated in the microwave at power six for 5 min, cooled for 20 min at room temperature, heated again at power six for 4 min, and cooled again for 20 min at room temperature. If there were vacant slots in the slide container, blank slides were used to ensure all slots were filled. Doing this helped facilitate a uniform distribution of the heat to each slide, which would in turn reduce differences in the staining intensity and tissue morphology due to uneven heating. With the PC protocol, slides were laid flat inside a microwavable PC and submerged in 0.01 M sodium citrate buffer (pH 6) and heated using a microwave at power 10 for 8 min. Slides were then cooled at room temperature for 45 min. Two enzymatic AR protocols were tested: 0.25% trypsin (1.1 mM in PBS) (Corning) and proteinase K (20 µg/mL in PBS) (Thermo Fisher, Waltham, MA, USA). Tissue sections were encircled with an ImmunoPen (Millipore Sigma, Burlington, MA, USA) and incubated with proteinase K or trypsin for 10 min at 37 °C. In the fifth condition, no AR was performed. Upon optimization, HIER with the microwave was used for all placental samples and for staining HLA-E in UC samples. Enzymatic AR with proteinase K was used for staining HLA-ABC and HLA-G in UC samples.

Following AR, slides were washed twice in PBS for 5 min each and incubated in 3% hydrogen peroxide in methanol for 30 min at room temperature to quench the endogenous peroxidase activity in the tissue. Slides were washed twice in PBS for 5 min each, and non-specific binding was blocked by incubating slides in 5% normal goat serum (Sigma Aldrich, St. Louis, MO, USA) in Tris-buffered saline with 0.1% Tween 20 (GE Healthcare, Chicago, IL, USA) (*v*/*v*) (TBS-T) for 1 h at room temperature. Without washing, one tissue section of each slide was incubated with the primary antibody, and the other was incubated with the mouse IgG1 antibody as the negative isotype control (Clone G3A1; Cat 5415S, Cell Signalling) at 4 °C overnight. The primary antibodies used were a mouse monoclonal anti-human pan HLA-ABC antibody at 0.25 µg/mL (Clone EMR8-5; Cat 565292, BD Biosciences, Franklin Lakes, NJ, USA), two different mouse monoclonal anti-human HLA-G antibodies at 2.0 µg/mL (Clone MEM-G/1; Cat MA1-19219, Thermo Fisher) (Clone 4H84; Cat 11-499-C100, Exbio, Vestec, Czechia), and a mouse monoclonal anti-human HLA-E at 5.0 µg/mL (Clone MEM-E/02; Cat ab2216, Abcam, Cambridge, UK). The negative isotype controls were used at the same working concentration as corresponding primary antibodies. After washing the slides twice in PBS for 5 min each, slides were incubated with a goat anti-mouse biotin secondary antibody at 2.6 µg/mL (Cat 31800, Thermo Fisher) for 1 h at room temperature. For both primary and secondary antibodies, 5% normal goat serum in TBS-T was used as the diluent. After two 5 min PBS washes, slides were incubated with the VECTASTAIN Elite Avidin Biotin Complex Peroxidase Kit (Cat VECTPK6100, Vector Laboratories, Newark, CA, USA) for 30 min and developed with a 3,3′-diaminobenzidine substrate kit (Cat SK-4100, Vector Laboratories) for 5 min at room temperature. Slides were washed once in dH_2_O for 3 min, counterstained with the Gill No. 2 Hematoxylin (Millipore Sigma) for 30 s, and washed in warm running tap water for 5 min. Finally, slides were dehydrated in 70%, 90%, and 100% ethanol, cleared in xylene, mounted with Cytoseal 60 (Richard-Allan Scientific, Kalamazoo, MI, USA), and imaged with the TissueScope LE Slide Scanner (Huron, Chicago, IL, USA). Staining results were graded based on staining intensity as follows: 0 (no staining or distinct differences from the isotype control), + (weak brown staining), ++ (moderate brown staining), +++ (strong dark brown staining).

### 2.3. Hematoxylin and Eosin Stain

Hematoxylin and eosin (H&E) stains of UC and placental tissue sections were performed. Tissue sections were deparaffinized in xylene and rehydrated in a decreasing gradient of ethanol in dH_2_O. Slides were stained with Harris Hematoxylin (Leica, Wetzlar, Germany) for 6 min. Following one wash in tap water for 2 min, the slides were de-stained with 1% acid alcohol (Leica) for 6 s. Then, slides were washed in tap water for 4 min and dipped in 80% ethanol for 30 s. The slides were counterstained with eosin (Leica) for 15 s and were then dehydrated by being dipped once in 95% ethanol for 10 s and twice in 100% ethanol for 30 s each. Finally, slides were cleared in xylene and mounted with Cytoseal-XYL (Richard-Allan Scientific). Slides were imaged with the TissueScope LE Slide Scanner (Huron).

### 2.4. Tissue and Cell Lysate Preparation

The lysis buffer used for both tissue and cell lysates was radioimmunoprecipitation assay (RIPA) buffer (NaCl (150 mM), Nonidet P-40 (1%), sodium deoxycholate (0.5%), SDS (0.1%), and Tris (50 mM, pH 7.4)) containing cOmplete ULTRA protease inhibitor cocktail (Roche Diagnostics, Basel, Switzerland). Tissues stored at −80 °C were thawed and grinded in RIPA buffer on ice until they were mostly homogenized. Then, tissues were allowed to further digest in solution on ice for 1 h and centrifuged at 1000× *g* for 5 min at 4 °C to collect the tissue lysate supernatant. Cell lysates were collected as previously described [35]. Briefly, cells were collected in RIPA buffer, allowed to be further digested on ice for 1 h, then centrifuged at 1000× *g* for 5 min at 4 °C to remove insoluble cell debris.

### 2.5. Western Immunoblotting

Western blot analysis was performed as previously described using equal amounts of total protein from UC tissue lysates, with placental tissue lysates as positive controls and primary human microvascular endothelial cell (HMVEC) lysates as negative controls [36]. All washes and incubation periods were performed with gentle agitation at room temperature unless otherwise specified.

Briefly, 25 µg of total protein from lysates was added to the loading sample buffer with 5% β-mercaptoethanol (βME) (Millipore Sigma), heated at 95 °C for 5 min, then electrophoresed on a 10% SDS-PAGE gel. Then, proteins were transferred to polyvinylidene difluoride membranes (GE Healthcare), which were briefly soaked in methanol for 30 s and acclimated in TBS-T prior to protein transfer. Then, membranes were washed three times in TBS-T for 5 min each and non-specific binding was blocked by incubating membranes in 5% nonfat dry milk in TBS-T (*w*/*v*) for 1 h. Following three washes of TBS-T for 5 min each, membranes were incubated with a mouse monoclonal anti-human HLA-G antibody (Clone 87G; Cat 130-111-851, Miltenyi Biotec, Bergisch Gladbach, Germany) diluted to 1:500 (*v*/*v*) with TBS-T at 4 °C overnight. After three washes of TBS-T for 5 min each, membranes were incubated with a goat anti-mouse IgG secondary antibody (Cat ab205719, Abcam) diluted to 1:2000 (*v*/*v*) with 5% nonfat dry milk in TBST (*w*/*v*) for 1 h. The protein actin was used as the housekeeping protein and was probed on the same membranes that were used to detect HLA-G presence. The protocol used to probe for actin was the same as the one described for HLA-G unless otherwise specified. After the membrane was striped, non-specific binding was blocked again before primary antibody incubation. The primary antibody used was a polyclonal goat anti-actin antibody (clone I-19; cat. sc-1616, Santa Cruz, Santa Cruz, CA, USA) diluted to 1:2000 (*v*/*v*) with TBS-T, and the secondary antibody used was a polyclonal donkey anti-goat IgG antibody (cat. 705-035-147, Jackson ImmunoResearch Laboratories, West Grove, PA, USA). Finally, all immunoblots were visualized by enhanced chemiluminescence substrate Western Lightning Plus-ECL (Perkin Elmer, Waltham, MA, USA) and imaged using the Chemidoc Imaging System (Bio-Rad, Hercules, CA, USA).

### 2.6. Immunoprecipitation and Mass Spectrometry

Immunoprecipitation was performed to purify HLA-G molecules from UC tissue lysates prior to mass spectrometry (MS) analysis. All washes and incubation periods were performed on the rotator at 4 °C unless otherwise specified. For each lysate mixture, 20 µL of magnetic beads (BioLabs, Singapore) were aliquoted and washed three times with PBS for 3 min each. To attach the mouse monoclonal anti-human HLA-G antibody (Clone 87G; Cat 130-111-851, Miltenyi Biotec) to the beads, the antibody was diluted to 1:100 in PBS with 0.02% Tween 20 and incubated with the beads overnight. After removing the supernatant with a magnetic rack, 1 mg of total protein from UC tissue lysates was added to the antibody-attached beads and incubated overnight to purify HLA-G molecules from lysates. HLA-G molecules in the tissue lysates were now attached to the anti-HLA-G antibody on the magnetic beads. After removing the supernatant, the beads mixtures were first washed three times in RIPA buffer containing the cOmplete ULTRA protease inhibitor cocktail (Roche Diagnostics) for 10 min each, and they were then washed three times in Tris-EDTA buffer (Wisent Bio Products, Saint-Jean-Babtiste, QC, Canada) for 10 min each. After removing the buffer, 20 µL of the sample buffer with 10% βME was added to each beads mixture and heated at 95 °C for 10 min to elute the HLA-G-antibody complex from the beads. Then, the mixtures were briefly vortexed and centrifuged to collect the supernatant, which contained the purified HLA-G. They were then stored at −80 °C.

To prepare samples for the MS analysis, the denatured and purified protein samples were electrophoresed on a 10% SDS-PAGE gel. The gel was briefly washed with dH_2_O in a clean tray, then incubated in the Coomassie blue stain (Abcam) overnight with gentle agitation at room temperature. To wash out excess Coomassie blue, the gel was washed for 4 h in dH_2_O with a change of clean dH_2_O every hour. Finally, the band of interest was excised from the gel and analyzed by MS with a blank area of the gel as negative control. The resulting data were interpreted using the Biopharma Finder software (Thermo Fisher).

## 3. Results

### 3.1. Optimization of IHC Protocol in UC Tissue Cross-Sections

Since HLA-ABC is expressed in almost all nucleated cells in adults, it is expected to be also present in the placenta and UC. Therefore, the IHC protocol optimization of placental and UC samples was conducted based on the HLA-ABC staining results. The criteria used to select the optimal IHC protocol using HLA-ABC staining results included minimal physical damage to tissue sections during staining, a consistent staining intensity, and minimal non-specific binding.

To begin the optimization process, three different working concentrations of the anti-HLA-ABC antibody (0.25 µg/mL, 0.125 µg/mL, and 0.05 µg/mL) and five different AR protocols were tested using placental tissue sections (Appendix A). The AR protocols tested in the optimization process are described in three groups: group 1 included HIER with the microwave or PC, group 2 included enzymatic retrieval with trypsin or proteinase K, and group 3 was the omission of any AR treatment. As the primary antibody concentration decreased, HLA-ABC staining intensity weakened in groups 1 and 2 and became absent in group 3 (Appendix A). Placental sections stained with 0.25 µg/mL of the anti-HLA-ABC antibody had the highest staining intensity out of all working concentrations tested (Appendix A). Specifically, strong staining was noted in proteinase K-treated sections, moderate staining was noted in group 1 and trypsin-treated sections, and weak staining was noted in group 3 sections (Appendix A). Among the placental sections stained with a working concentration of 0.25 µg/mL, undesirable tissue damage was present in PC-treated sections and sections without AR (Appendix A). Non-specific binding was noted in group 2 placental sections (Appendix A). Based on these observations, HIER using the microwave was performed for all subsequent IHC stains of placental tissue sections. The anti-HLA-ABC antibody working concentration of 0.25 µg/mL was used for the IHC optimization of UC tissue sections and all subsequent IHC experiments. Finally, optimal working concentrations of the anti-HLA-G and anti-HLA-E primary antibodies were determined using term placental tissue sections. The IHC staining results of HLA-ABC, HLA-G, and HLA-E in placental tissue sections using the determined optimal primary antibody working concentrations are summarized in Figure 1.

To highlight the morphological differences within the WJ and offer further clarity when describing the IHC staining results of UC tissue cross-sections, we will refer to specific parts of the UC by the anatomical terminologies defined in a review from 2017 [16]. The amniotic epithelium (AE) is the most outer layer of the UC, and we observed it to be generally one-cell thick (Figure 2). The WJ can be divided into three subregions: the subamniotic WJ (SAWJ), the intermediate WJ (IWJ), and the perivascular WJ (PVWJ) (Figure 2). The SAWJ lies immediately beneath the AE. The PVWJ is at the center of the UC surrounding the umbilical vessels (Figure 2). The IWJ is located between the SAWJ and PVWJ, and it could be readily distinguished from the other two subregions by its loosely structured matrix (Figure 2). Visible clefts could be noted in the IWJ and were absent in both cells and WJ (Figure 2). Structurally, the PVWJ was the most compact among the WJ subregions, and cells in the WJ were most densely populated in the PVWJ (Figure 2). Finally, three umbilical vessels exist in the UC: one umbilical vein and two umbilical arteries (Figure 2). The umbilical vessels are comprised of only the tunica intima and media; the tunica externa is absent in all three vessels (Figure 2).

The optimal fixation conditions and AR protocol were determined for UC samples using UC tissue sections at term (Table 3). Due to its loose structure, the IWJ is the most fragile part of the UC during the AR step. Therefore, one of the main goals during the IHC optimization process was to maximize IWJ tissue preservation while maintaining a consistent staining intensity. Among the tested fixation conditions, UC tissues fixed for 48 h had an overall stronger staining intensity than those fixed for 24 h (Table 3). Within the 48 h group, UC tissues fixed with rotary agitation at room temperature was the only condition with low levels of both non-specific binding and tissue disruption (Table 3). Although the UC tissues fixed with rotary agitation at 4 °C had the most consistent staining intensity with low tissue disruption, more non-specific binding was noted compared with those fixed with rotary agitation at room temperature (Table 3). Therefore, the optimal fixation conditions were concluded to be 48 h with rotary agitation at room temperature.

The HLA-ABC IHC staining results of all tested AR protocols in UC tissue cross-sections at term can be found in Appendix A. Among the term UC tissue sections fixed with the abovementioned optimal conditions, tissue damage was present only in group 1. This was undesirable but expected, as the physical conditions of HIER protocols were the harshest out of all the AR protocols tested (Table 3). Though the IWJ naturally contains clefts void of cells and matrix, such areas in the group 1 UC sections were generally larger, and their boundaries had sharp edges with strands of tissue residue, which are indications of tissue loss and damage (Appendix A). Within group 1, PC-treated UC tissue sections had more severe tissue disruption than microwave-treated sections (Appendix A). Whereas tissue damage was generally limited to only the IWJ of microwave-treated sections, visible tissue loss and damage of the WJ matrix were noted in the IWJ and SAWJ of PC-treated sections (Appendix A). No tissue disruption was noted in groups 2 or 3 (Table 3).

Across all UC regions, group 2 had the strongest staining intensity (Table 3). The staining intensity of group 3 was the weakest at the various UC regions, particularly the AE and WJ (Table 3). The staining intensity of group 1 at the AE and WJ was comparable to group 2, but the umbilical vessels of group 1 tissue sections showed weaker staining than group 2 (Appendix A). Due to the presence of undesirable tissue damage in group 1 and weak staining in group 3, we compared the UC staining results within group 2 further to determine the optimal AR protocol. At the AE and endothelium of group 2 tissue sections, the staining intensity of proteinase K-treated UC tissue sections were scored as strong, whereas trypsin-treated sections were scored as moderate (Table 3). The WJ and vessel walls of proteinase K- and trypsin-treated sections were both scored as moderate (Table 3). The non-specific binding noted in the AE of proteinase K-treated sections was only present in one term UC sample and was absent in all subsequent HLA-ABC IHC stains, while the staining intensity at the AR remained strong. Considering the degree of tissue disruption, the prevalence of non-specific binding, and the staining intensity, enzymatic retrieval using proteinase K was deemed as the preferred AR method when performing IHC staining of whole UC tissue. Though the tissue disruption was less severe in microwave-treated sections, the IWJ was nonetheless better preserved in proteinase K-treated sections (Figure 2).

### 3.2. Localization and Levels of Class I HLA Molecules in UC

The IHC staining results of HLA-ABC, HLA-G, and HLA-E in UC tissue cross-sections were scored based on their staining intensities and irrespective of the prevalence of positive staining. Appendix A summarize the staining intensity scores of UC tissues at term and between 12 and 19 weeks, respectively. HLA presence in the WJ was mainly limited to cells within the collagenous matrix. The staining intensity of HLA-positive (HLA+) cells was constant across WJ subregions of UC tissue cross-section at both gestational ages. Similar observations were made for the endothelium and vessel walls of the umbilical vessels in each sample. Therefore, staining intensity scores of the WJ represent the intensity across WJ subregions, and scores given to the endothelium and vessel walls of each sample represent all three umbilical vessels of that sample. HLA-G was absent in all UC tissues analyzed (term and 12–19 weeks), as indicated by the lack of positive staining compared to the negative control (Figure 3, Figure 4, Figure 5 and Figure 6). Thus, the following analysis focused on the localization and levels of HLA-ABC and HLA-E.

The localization of HLA-ABC and HLA-E at the AE were similar among UC tissues at term and between 12and 19 weeks (Figure 3), but their levels differed. HLA-ABC was present in all UC tissues analyzed. HLA-E was present at the AE in nine out of ten UCs at term and eight out of ten UCs in early pregnancy. At both gestational ages, HLA-ABC levels were the highest, with a moderate to strong staining intensity, whereas HLA-E levels were considerably lower as shown by the weak staining intensity (Figure 3).

The positive staining of HLA-ABC and HLA-E was noted specifically in cells of the WJ subregions at both gestational ages, though not all WJ cells showed positive staining (Figure 3, Figure 4 and Figure 5). HLA-ABC+ cells in the WJ of UC tissues at term, especially those in the SAWJ and IWJ, had an overall weaker staining intensity compared with HLA-ABC at the AE (Figure 3 and Figure 4). This indicated a lower HLA-ABC level in the WJ than the AE. A greater portion of cells in the PVWJ were HLA-ABC+ compared with cells in the SAWJ and IWJ (Figure 5). A similar distribution of HLA-E+ cells across WJ subregions of UCs at term was noted, but their staining intensity remained weak (Figure 3, Figure 4 and Figure 5). Because the structural organization of the WJ subregions in UCs at 12–19 weeks is not as distinct as UCs at term, classification of the WJ subregions was based on the physical proximity of a given region to the AE or umbilical vessels. The WJ cells of UC tissues at 12–19 weeks had similar patterns of HLA localization and HLA levels as UCs at term (Figure 3, Figure 4 and Figure 5). Compared with HLA-ABC in the AE, the HLA-ABC level in the WJ was lower, with HLA-ABC+ cells showing a weak to moderate staining intensity in most UC tissues analyzed (Appendix A). Whereas HLA-ABC+ cells were noted in all WJ subregions, HLA-E tended to be more prevalent in the IWJ and PVWJ. Specifically, HLA-E presence in the SAWJ was only noted in half of the UC tissues analyzed (Appendix A). Overall, WJ subregions in UC tissues at term and 12–19 weeks had a higher HLA-ABC level than HLA-E.

HLA-ABC and HLA-E had similar localization patterns in the umbilical vessels of UC tissues at term. A moderate to strong staining intensity was observed in the endothelial cells after staining for either HLA-ABC or HLA-E (Appendix A). Compared to their levels at the endothelium, the vessel walls had lower HLA-ABC and HLA-E, as indicated by their staining intensity scores (Appendix A). Positive HLA-ABC staining could be noted both in the smooth muscle cells and at the intercellular spaces within the tunica media (Figure 6). On the other hand, HLA-E presence was mainly limited to within the smooth muscle cells, and its staining intensity was visibly lower than HLA-ABC (Figure 6). For UC tissues at term, the endothelium showed similar levels of HLA-ABC and HLA-E, while the vessel walls were higher in HLA-ABC than HLA-E (Figure 6).

Overall, for UC tissues between 12 and 19 weeks, HLA-ABC levels were higher than HLA-E at both the endothelium and vessel walls (Appendix A). At the endothelium, a moderate to strong staining intensity was noted for HLA-ABC, whereas the HLA-E staining intensity was weak to moderate (Appendix A). At the vessel walls, HLA-ABC levels were distinctly higher than HLA-E (Figure 6). Specifically, HLA-ABC was present in all UC samples analyzed with a weak to moderate staining intensity, whereas HLA-E was present in six out of ten samples analyzed with a weak staining intensity (Appendix A). In conclusion, HLA-ABC levels were higher than HLA-E at the umbilical vessels of UCs at both gestational ages, except at the endothelium of UCs at term.

A final comparison was made on the localization and levels of HLA-ABC and HLA-E between UCs at term and at 12–19 weeks. HLA-ABC and -E had similar localizations in most UC samples analyzed, regardless of gestational age (Appendix A). HLA-ABC was present in all parts of the UC, but it was higher at the AE and endothelium than at the WJ and vessel walls at both gestational ages (Appendix A). UCs at term and between 12 and 19 weeks shared similar patterns of HLA-E localization and level. HLA-E was present in all parts of the UC, among which the endothelium had the highest HLA-E staining intensity, and the vessel walls had the lowest (Appendix A). The only distinct difference in HLA-E levels between UCs at term and 12–19 weeks was that HLA-E had a moderate to strong staining intensity at the endothelium of UCs at term, whereas the endothelium of UCs between 12 and 19 weeks showed a weak to moderate staining intensity (Appendix A). For UC tissues at both gestational ages, HLA-ABC levels were higher than HLA-E.

### 3.3. HLA-G Complex in Whole UC Tissue

Additional experiments were performed to further investigate the unexpected lack of positive HLA-G IHC staining in all UC tissues analyzed. Western immunoblotting was performed on whole UC tissue lysates to determine HLA-G presence in the UC (Figure 7). The dominant band for the positive control and all UC tissue lysate samples was at 55 kDa (Figure 7). A trend of increasing HLA-G level with increasing gestational age was noted after a comparison of the intensity of the 55 kDa band between samples from the second trimester and samples at term (Figure 7).

To further confirm the presence of HLA-G in UC, we analyzed the 55 kDa dominant band in whole UC tissue at term using MS. HLA-G molecules from the UC tissue lysate was affinity purified by immunoprecipitation, and the 55 kDa band was excised from the Coomassie blue-stained gel. From the same gel, a blank area was excised and used as a negative control. The MS analysis revealed the presence of the HLA-G isoform G5 in the 55 kDa band (Figure 7). Specifically, there were 11 MS peaks with an HLA-G5 sequence coverage of 77.1%, leading to a 13.79% abundance of HLA-G5 (Figure 7). The peptide alignment maps of the band of interest and the negative control to the HLA-G5 sequence can be found in Appendix A.

## 4. Discussion

We sought to determine the localization and abundance of various class I HLA molecules to understand the immunogenic and immunosuppressive profiles of the UC. We observed HLA-ABC to be present throughout the UC, with its highest abundance in the AE and endothelium. We observed minor levels of the immunomodulatory HLA-E within the UC, and most of it was present in the AE, WJ, and the endothelium. HLA-E was not consistently present in all patient UC samples. The IHC staining of the UC consistently showed an absence of HLA-G regardless of gestational age, but further investigation by Western immunoblot and MS analyses revealed a presence of the HLA-G5 isoform at low abundance in UC tissues at term.

To our knowledge, this study is the first to understand the relative localization and levels of these important immune-regulatory molecules in whole UC tissue. Upon initial examination, UCs from first-trimester patients revealed very poor structural integrity. The physical characteristics of first-trimester UCs steered our efforts towards second- and third-trimester UCs since future applications would require UC lumens to be physically manipulatable. Overall, we observed higher levels of HLA-E staining intensity in UC tissues at term compared with those in the second trimester, while HLA-ABC levels were similar in both gestational ages. Regardless of gestational age, we did not observe any positive HLA-G staining using IHC in any of the UC tissues.

Based on our findings, discordance exists regarding the presence of HLA-G in the UC. Two anti-HLA-G antibody clones were used in the IHC staining of UC tissues: 4H84 and MEM-G/1. HLA-G5 was detected to be present in the UC tissue by Western immunoblot and MS analyses, but no positive HLA-G staining was observed by IHC. In contrast, both 4H84 and MEM-G/1 consistently detected HLA-G presence in placental tissues in early pregnancy and at term, which were used as positive controls. This shows that the reliability and sensitivity of 4H84 and MEM-G/1 are sufficient for HLA-G detection in placental tissues but not in UC tissues where the HLA-G level is likely low as we observed. A similar lack of concordance was noted in the detection of HLA-G in rectal cancer tissue [37]. Specifically, HLA-G detection by IHC using 4H84 and MEM-G/1 were similarly consistent in placental tissues used as positive controls, but the staining in rectal cancer tissue was inconsistent and in conflict with the Western immunoblot analysis [37]. In conjunction with the discordance in the present study, caution should be taken when interpreting the detection of HLA-G in non-placental tissues using the anti-HLA-G 4H84 and MEM-G/1 antibodies.

Based on the observations reported in the current study, HLA-G may not be the main contributor of the immunosuppressive properties observed in UC allografts. Furthermore, a recent study reported an absence of HLA-G expression and high levels of HLA-ABC expression in UCMSCs using 10x single-cell RNA sequencing analysis [18]. Though the study was only in vitro and was limited in its sample size, its reported findings nonetheless support the notion that additional contributors to immunosuppression exist in the UC tissue. Based on our observations, the low immunogenicity of UC allografts may be more due to the interaction between HLA-ABC and HLA-E and the effect of this interaction on the immune cells. Considering that HLA-G was found to be present at low levels in the whole UC tissue whereas HLA-ABC and HLA-E were much more prevalent, the HLA compatibility of the UC allograft to its recipient should be considered on a per-patient basis.

To our knowledge, our IHC results demonstrated for the first time that the presence of HLA-E in the UC should be considered when building the UC as a potential graft source for transplantation in urologic reconstruction. Our observation that there seems to be a paired expression of HLA-ABC and HLA-E at the same regions within the UC tissue is promising despite the lack of HLA-G expression. Normally, HLA-E is known to present peptides derived from the leader sequences of HLA-ABC [33,34,38]. Our observations of similar HLA-ABC and HLA-E localization patterns across UC tissues are consistent with the reported positive correlation between HLA-E surface expression and HLA-ABC leader peptide (LP) levels that bind to HLA-E [39,40]. By presenting LPs from HLA-ABC, HLA-E interacts with the inhibitory CD94/NKG2A receptors at greater affinity than the activating CD94/NKG2C receptors on NK cells [41]. The NK cells thus receive a net inhibitory signal, and the HLA-E^+^ cells are protected from NK cell lysis [41,42].

In the context of allogeneic graft transplants, mixed results have been reported regarding the effect of HLA-E on transplant outcome and graft survival in clinical and animal studies. HLA-E mismatch between the donor and recipient patients with acute leukemia was found to be associated with a better five-year overall survival and lower graft vs. host disease incidence in the recipient patients [43]. In another study, the renal allograft biopsies of patients with acute cellular rejection were found to have higher levels of HLA-E and were associated with a greater number of LP mismatches between the allograft and the recipient cells [44]. When studying the effect of the Qa-1, which is the mouse homolog of HLA-E, on allotransplant outcome, the production of allograft-specific antibodies and allograft rejection were found to be reduced by Qa-1-specific CD8 regulatory T cells [45]. Heterogeneity exists in the effect of HLA-E on allotransplantation outcomes, and this should be taken into consideration when the UC is used as an allograft.

The UC has been used as allografts in the repair of gastroschisis, foot and ankle wounds, and diabetic foot ulcers [12,13,14,15]. In one pilot study, the UC allograft was used to wrap around the repaired tendon, and no adverse events occurred after the surgical repair in all five patients [12]. In fact, there was less post-operative pain and inflammation, and patients returned to normal activities much sooner [12]. The UC has also been used to cover the exposed intestine in neonatal patients with gastroschisis when the conventional method to close the defect was not possible [13]. Of the 24 out of 27 patients that survived, 16 had complete healing of the gastroschisis defect without needing further operations [13]. Similar wound healing properties were observed in the treatment of complex diabetic foot ulcers [14,15]. The healing rate after UC allograft application was higher compared with the healing rate following standard care, and 86% of patients had confirmed wound healing in the one-year follow-up study [14,15]. Based on the encouraging results of available applications of the UC allograft, we propose that the UC allograft can also be used for urethral reconstruction.

In severe cases of hypospadias, the buccal mucosal graft or preputial skin graft are commonly used for urethral reconstruction [46]. However, re-operations due to complications such as urethral stricture are a frequent occurrence, and the risk for additional operations increases with each re-operation [4]. Whereas autologous grafts taken from the buccal mucosa or prepuce have limited graft availability, the UC is an inexhaustible graft source and likely has immunomodulatory properties, as suggested by our findings. However, challenges exist with this proposed application. The first challenge is that the UC tissue needs to differentiate towards a urethra-like morphology despite the human urethra being structurally more complex than the UC [3]. Considering that the UC tissue graft has been previously shown to develop a similar morphology as the repairment site, its differentiation to a urethra-like morphology is possible. However, additional physical stimuli may be required for the graft to be mechanistically similar to the urethra [47]. Another challenge is the adequate and efficient vascularization of the graft following transplantation, which directly impacts the graft outcomes. The composition of the UC WJ includes growth factors such as the vascular endothelial growth factor and cytokines associated with wound healing, which can aid in neovascularization and tissue regeneration [17]. In consideration of the identified challenges, further studies are necessary to evaluate the efficacy of the UC graft for the purposes of urethral reconstruction.

Limitations exist in the present study. Since the sample size of this study was small, the reported findings should be used as the initial supporting evidence for more extensive research on the clinical implications of the immunological profile of UC allografts. Using a preclinical model, it would be important to investigate whether the localization and relative levels of HLA molecules in different UC regions contribute to differences in the engraftment outcome. Since the present study focused on the semi-quantitative comparison between HLA molecules by histological staining, future quantitative analyses are needed to further confirm the suggested immunological interactions between HLA-ABC and HLA-E. It would be useful to sort hematopoietic progenitor cell subpopulations by their HLA expression using flow cytometry. Furthermore, the differences in their long-term culture-initiating cell (LTC-IC) frequency and generation of more mature precursor cells, such as burst forming unit-erythroid (BFU-E) cells (BFU-E), could be explored in colony and co-culture assays. In conjunction with the findings reported in this study, gaining a deeper understanding of the hematopoietic cell subpopulations in the UC tissue will further elucidate mechanisms behind the UC’s low immunogenicity in allogeneic environments.

Overall, we observed a paired localization of HLA-ABC and HLA-E in UC tissue and that the level of HLA-G appears to be lower than HLA-ABC and HLA-E. It is likely that the UC maintains some level of immunosuppression by virtue of the paired expressions of HLA-ABC and HLA-E and their known interactions on immune cell activity. Future attempts to develop tissue grafts out of UC should use the histology protocol we described to characterize and determine the utility of any UC that is to be developed into a future graft.

## Figures and Tables

**Figure 1 bioengineering-10-00110-f001:**
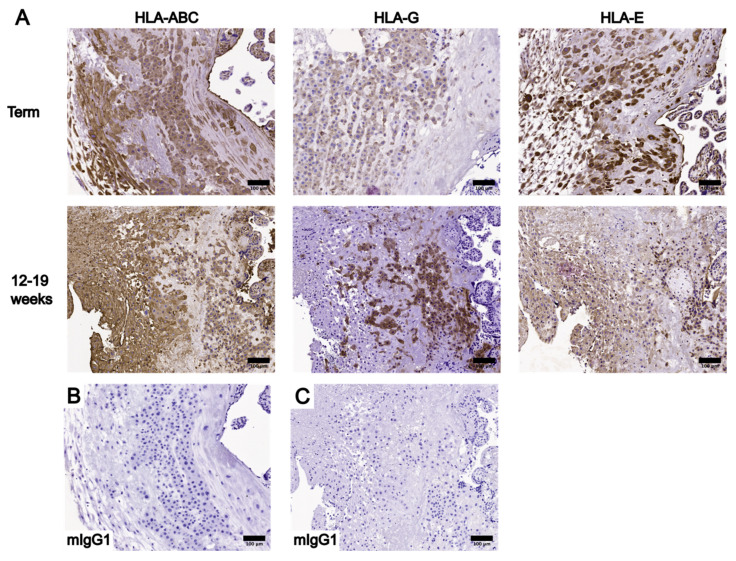
Summary of the immunohistochemstry (IHC) staining results of HLA-ABC, HLA-G, and HLA-E in placental tissue sections at the decidua using the optimized primary antibody working concentrations. (**A**) Placental tissue sections at term and between 12 and 19 weeks stained with HLA-ABC (0.25 µg/mL), HLA-G (2.0µg/mL), and HLA-E (5.0 µg/mL). (**B**) Negative control of term placental IHC staining results using an anti-mouse IgG1 (mIgG1) antibody. (**C**) Negative control of 12–19 week placental IHC staining results using an anti-mIgG1 antibody. Scale bars = 100 µm.

**Figure 2 bioengineering-10-00110-f002:**
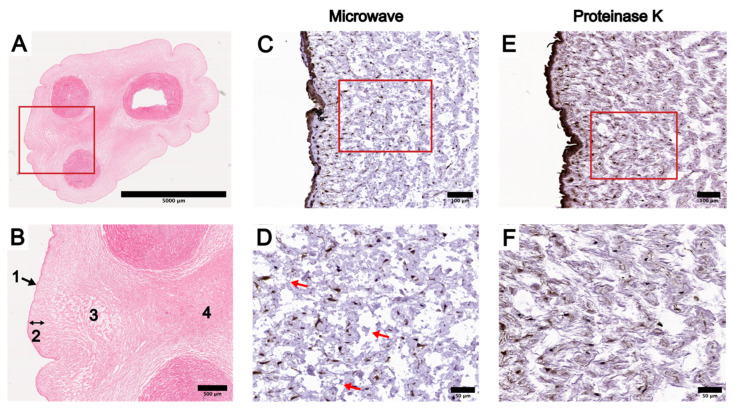
Comparison of HLA-ABC staining results in term umbilical cord (UC) cross-sections using microwave and proteinase K antigen retrieval (AR) protocols. (**A**) Hematoxylin and eosin stain of a term UC cross-section showing the outer amniotic epithelium (AE), three umbilical vessels, and the Wharton’s Jelly (WJ) surrounding the vessels. (**B**) Enlarged image of the boxed region in (**A**): 1, AE; 2, subamniotic WJ (SAWJ) located between the AE and the intermediate WJ (IWJ); 3, more loosely structured IWJ with visible clefts in the matrix; 4, dense perivascular WJ (PVWJ) between the two vessels where the matrix structure is more compact. (**C**) IHC staining of HLA-ABC in term UC cross-sections using the microwave AR protocol, focusing on the AE and SAWJ. (**D**) Boxed region of (**C**) at higher magnification. Red arrows: visible disruption of IWJ after microwave AR in comparison to proteinase K AR. (**E**) IHC staining of HLA-ABC using the proteinase K AR protocol in term UC cross-sections, focusing on the AE and SAWJ. (**F**) Boxed region of (**E**) at higher magnification. Scale bars: A = 5000 µm; B = 500 µm; C and E = 100 µm; D and F = 50 µm.

**Figure 3 bioengineering-10-00110-f003:**
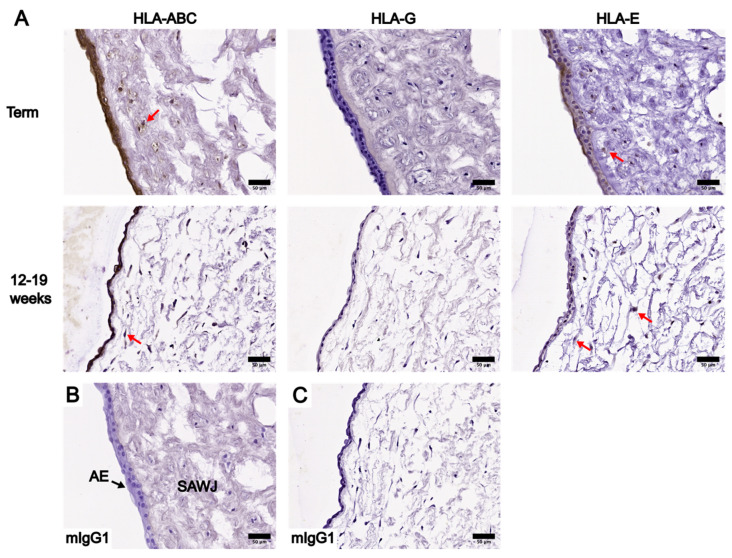
Comparison of IHC staining results of HLA-ABC, HLA-G, and HLA-E in whole UC cross-sections at term and between 12 and 19 weeks, focusing on the AE and SAWJ. (**A**) IHC staining results of UC cross-sections at term are shown in the first row, and IHC staining results of UC between 12 and 19 weeks are shown in the second row. Red arrows: examples of cells in the SAWJ showing positive staining for the corresponding HLA of interest. (**B**) Negative control of term UC IHC staining results using an anti-mIgG1 antibody. (**C**) Negative control of 12–19-week UC IHC staining results using an anti-mIgG1 antibody. Scale bars = 50 µm.

**Figure 4 bioengineering-10-00110-f004:**
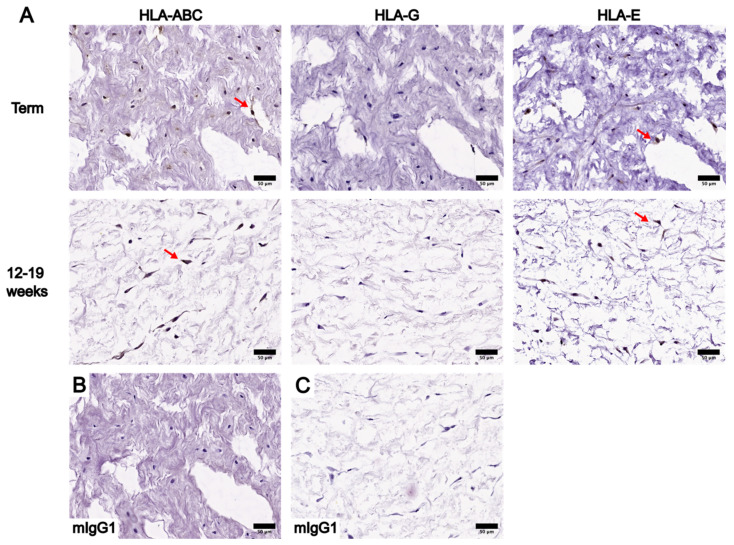
Comparison of IHC staining results of HLA-ABC, HLA-G, and HLA-E in whole UC cross-sections at term and in 12–19 weeks, focusing on the IWJ. (**A**) IHC staining results of term UC are shown in the first row, and IHC staining results of UC between 12 and 19 weeks are shown in the second row. Red arrows: examples of cells in the IWJ showing positive staining for the corresponding HLA of interest. (**B**) Negative control of term UC IHC staining results using an anti-mIgG1 antibody. (**C**) Negative control of 12–19-week UC IHC staining results using an anti-mIgG1 antibody. Scale bars = 50 µm.

**Figure 5 bioengineering-10-00110-f005:**
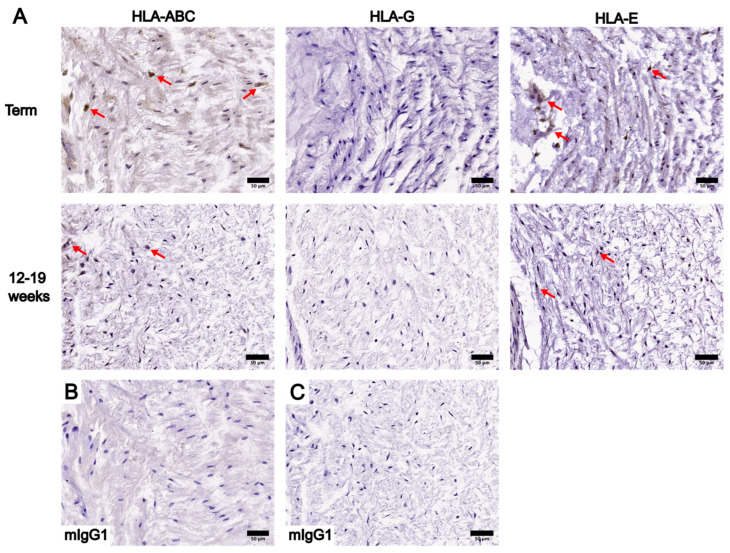
Comparison of IHC staining results of HLA-ABC, HLA-G, and HLA-E in whole UC cross-sections at term and between 12 and 19 weeks, focusing on the PVWJ. (**A**) IHC staining results of term UC are shown in the first row, and IHC staining results of UC between 12 and 19 weeks are shown in the second row. Red arrows: examples of cells in the PVWJ showing positive staining for the corresponding HLA of interest. (**B**) Negative control of term UC IHC staining results using an anti-mIgG1 antibody. (**C**) Negative control of 12–19-week UC IHC staining results using an anti-mIgG1 antibody. Scale bars = 50 µm.

**Figure 6 bioengineering-10-00110-f006:**
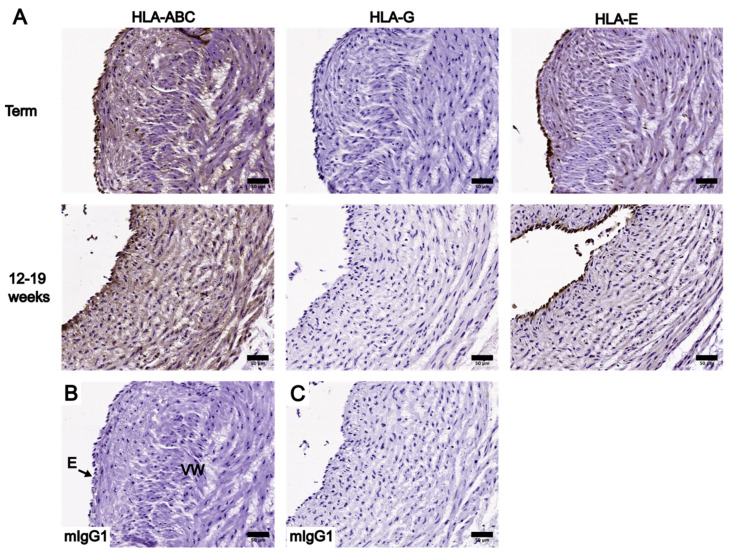
Comparison of IHC staining results of HLA-ABC, HLA-G, and HLA-E in whole UC cross-sections at term and between 12 and 19 weeks, focusing on one of the umbilical vessels. (**A**) IHC staining results of term UC are shown in the first row, and IHC staining results of UC between 12 and 19 weeks are shown in the second row. (**B**) Negative control of term UC IHC staining results using an anti-mIgG1 antibody. (**C**) Negative control of 12–19-week UC IHC staining results using an ant-mIgG1 antibody. Scale bars = 50 µm.

**Figure 7 bioengineering-10-00110-f007:**
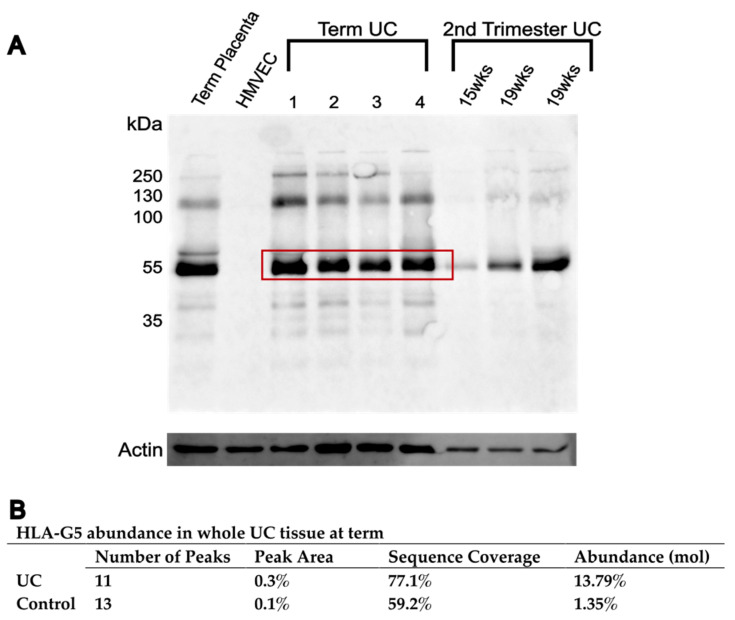
Investigation of HLA-G presence in whole UC tissue. (**A**) Western immunoblotting of whole UC tissue lysate using samples at term (*n* = 4) and in early pregnancy (*n* = 3). Term placental tissue lysate was used as the positive control, and primary human microvascular endothelial cell (HMVEC) lysate was used as the negative control. Actin was used as the housekeeping protein. (**B**) Mass spectrometry analysis of HLA-G in a whole UC tissue at term (from boxed region in (**A**)) revealed a low abundance of HLA-G5. A blank area of the gel was used as negative control.

**Table 1 bioengineering-10-00110-t001:** Clinical parameters of participants.

	Term
n	10
Maternal age (year)	34.6 ± 2.1
Gestational age (weeks)	38.8 ± 0.3
Body mass index (kg/m^2^)	32 ± 2.9
Gravidity	3 ± 0.6
Parity	1 ± 0.3
Blood pressure (mmHg)	
Systolic	109 ± 1.8
Diastolic	65 ± 1.5
Proteinuria (plus protein)	0 ± 0
Birth weight (g)	3594 ± 158.7

Values are mean ± Standard error of mean.

**Table 2 bioengineering-10-00110-t002:** Primary antibodies used in IHC stains.

Antibody	Clone	Species	Working Concentration (µg/mL)	Source
HLA-ABC	EMR8-5	Mouse	0.25	BD Biosciences (cat 565292)
HLA-G	MEM-G/1	Mouse	2.0	Thermo Fisher (MA1-19219)
	4H84	Mouse	2.0	Exbio (cat 11-499-C100)
HLA-E	MEM-E/02	Mouse	5.0	Abcam (cat ab2216)

**Table 3 bioengineering-10-00110-t003:** Comparison of HLA-ABC IHC staining results under different fixation conditions using five different AR protocols in term UC cross-sections.

AR		Fixation Condition							
		Fixed for 24 h				Fixed for 48 h			
		With Agitation		No Agitation		With Agitation		No Agitation	
		4C	RT	4C	RT	4C	RT	4C	RT
Mic	AE	++#	++#	++#	++#	++	++#	++	++#
	WJ *	++ (Present)	++# (Present)	++# (Present)	++# (Present)	++# (Present)	++# (Present)	++# (Present)	++ (Present)
	E	++	++	++	++	++	++	++	++
	VW	+	+#	+#	+#	+#	+#	+#	+#
PC	AE	++	++	++	++	++#	++	++	++#
	WJ*	++ (Present)	++# (Present)	++# (Present)	++ (Present)	++ (Present)	++# (Present)	++# (Present)	++ (Present)
	E	++	++	++	++	++	++	++	++
	VW	+#	++#	++	++#	++#	++#	+#	+#
Trypsin	AE	+++#	++#	++#	+++#	+++#	++	+++#	++#
	WJ *	++# (Absent)	++# (Absent)	++ (Absent)	++ (Absent)	++# (Absent)	++ (Absent)	++ (Absent)	++ (Absent)
	E	++~+++	+	++	++	++	++	+++	++
	VW	++#	+	+#	+	++#	++	++#	++
ProK	AE	+++	+++#	+++	+++#	+++#	+++#	+++#	+++#
	WJ *	++# (Absent)	++ (Absent)	++ (Absent)	++ (Absent)	++ (Absent)	++ (Absent)	++ (Absent)	++ (Present)
	E	++	++	++	++	++	+++	+++#	++
	VW	++	++	++	+	++	++	++#	++
No AR	AE	++	++	++	++	++	++#	+~++	++#
	WJ *	+ (Absent)	+ (Absent)	+++ (Absent)	++ (Absent)	++ (Absent)	+ (Absent)	++# (Absent)	++ (Absent)
	E	+	+	+	++	++	++	+	++
	VW	+	+	+	+	++	+	+	+

* An overall assessment of all three parts of the WJ. Data are presented as staining intensity scores (presence/absence of any tissue disruption). # Presence of non-specific binding. Abbreviations: amniotic epithelium (AE); endothelium (E); microwave (Mic); pressure cooker (PC); proteinase K (ProK); vessel wall (VW); Wharton’s Jelly (WJ). Staining intensities were graded as such: 0 (no staining or distinct differences from the isotype control), + (weak brown staining), ++ (moderate brown staining), +++ (strong dark brown staining).

## Data Availability

The data presented in this study are available in the present article or in Appendix A.

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
