# Peer review of "Histological Characterization of Class I HLA Molecules in Whole Umbilical Cord Tissue Towards an Inexhaustible Graft Alternative for Reconstructive Surgery"

_bioengineering, 2023, doi:10.3390/bioengineering10010110_

Round 1
Reviewer 1 Report
In this study, the authors tried to explore the histological characterization of Class I HLA molecules in whole umbilical cord tissue towards an graft alternative for reconstructive surgery. It's a meaningful study and does provide some new ideas to the readers. The manuscript is well-written and easy to read. The only problem is the lack of in vivo studies. The authors should explain this. So, minor revision should be recommended for this manuscript.
Author Response
Dear Reviewer 1,
Thank you for your time and effort in reviewing our manuscript. We appreciate your comments and have attached our point-by-point response.

Reviewer 2 Report
The article "Histological Characterization of Class I HLA molecules in 2 Whole Umbilical Cord Tissue Towards an Inexhaustible Graft Alternative for Reconstructive Surgery" by Yue Ying Yao Dennis K. Lee , Stephanie Jarvi, Marjan Farshadi, Minzhi Sheng, Sara Mar and Ori Nevo Hon S. Leong investigated and compared the localization and levels of HLA-ABC, -G, and -E in whole-cord cross-sections to better understand the allogenicity of the UC. The article is well planned with well-done experiments, however I suggest the authors to implement the quality of their manuscript with a flow cytometric analysis of the CD34 cell population positive for HLA-ABC, -G and E. In this way it is possible to define the quantitative and qualitative relationships between the different cells' subpolations more specifically.
In addition,I have two more questions for the authors:
Do CD34+ HLA-ABC+, -G+ and E+ cells generate BFU-E, CFU-GM and CFU-GEMM with a different frequency than CD34+ HLA-ABC-, -G- and E- cells? the Authors could perform a colony assay in methylcellulose.
Does the frequency of LTC-IC change? the authors could sort CD34+ HLA-ABC+, -G+ and E+ and perform a co-culture assay on fibroblast monolayer.
Author Response
Dear Reviewer 2,
Thank you for your time and effort in reviewing our manuscript. We appreciate your comments and have attached our point-by-point response.

Reviewer 3 Report
In the paper titled “Histological Characterization of Class I HLA molecules in Whole Umbilical Cord Tissue Towards an Inexhaustible Graft Alternative for Reconstructive Surgery” the authors investigate and compare the localization and levels of HLA-ABC, -G, and -E in whole-cord cross-sections to better understand the allogenicity of the UC.
I thing that this work is interesting but it needs of a major revision:
My observations are as follows:
· I propose to add into the work the meaning of gravidity and parity, which are the number of times a woman is or has been pregnant (gravidity) and has carried pregnancies to a viable gestational age (parity), as not everyone is familiar with these terms
· Check the English form
· Clarify microwave use and exclude confounding factors arising from microwave use
· Be very cautious about the results since there are few samples
· Why is there a difference in the intensity of the 55-kDa band in the 2 samples at week 19? Perhaps it is better to make the caption of Figure 7 clearer
· Has a statistical analysis of the results been done?
· Better argue on the possible applications in humans and any limitations
Author Response
Dear Reviewer 3,
Thank you for your time and efforts in reviewing our manuscript. We appreciate your comments and have attached a point-by-point response.

Round 2
Reviewer 2 Report
The authors did not perform any of the requested experiments. they only reported in the text that further molecular investigations are needed. Considering that the work has a strong histological focus it could also be accepted, even if I believe that some molecular aspects would have enhanced it more.Author Response
Dear Reviewer 2,
Thank you for your time and effort in reviewing our manuscript again, we have provided a point-by-point response to your comment. We have not made any new changes in the newly uploaded manuscript.
Point 1: The authors did not perform any of the requested experiments. they only reported in the text that further molecular investigations are needed. Considering that the work has a strong histological focus it could also be accepted, even if I believe that some molecular aspects would have enhanced it more
Response 1: We did not, and this is due to the strong histologic focus of the paper. We agree with Reviewer #2 that some molecular aspects will need to be investigated and we hope to do this in the future since. This will require acquisition of new material from patients to perform the functional assays and flow cytometry experiments.
Reviewer 3 Report
Accept in present form
Author Response
Dear Reviewer 3,
Thank you for your time and effort in reviewing our manuscript again, we have attached our response to your comments.
